# Knowledge about Snake Venoms and Toxins from Colombia: A Systematic Review

**DOI:** 10.3390/toxins15110658

**Published:** 2023-11-15

**Authors:** Jaime Andrés Pereañez, Lina María Preciado, Paola Rey-Suárez

**Affiliations:** 1Research Group in Toxinology, Pharmaceutical, and Food Alternatives, Pharmaceutical and Food Sciences Faculty, University of Antioquia, Medellín 50010, Colombia; maria.preciado@udea.edu.co (L.M.P.); jessica.rey@udea.edu.co (P.R.-S.); 2Research Group in Pharmaceutical Promotion and Prevention, University of Antioquia, Medellín 50010, Colombia; 3Centro de Investigación en Recursos Naturales y Sustentabilidad, Universidad Bernardo O’Higgins, Santiago 8320000, Chile

**Keywords:** Colombia, proteomic, snake venom, toxin

## Abstract

Colombia encompasses three mountain ranges that divide the country into five natural regions: Andes, Pacific, Caribbean, Amazon, and Orinoquia. These regions offer an impressive range of climates, altitudes, and landscapes, which lead to a high snake biodiversity. Of the almost 300 snake species reported in Colombia, nearly 50 are categorized as venomous. This high diversity of species contrasts with the small number of studies to characterize their venom compositions and natural history in the different ecoregions. This work reviews the available information about the venom composition, isolated toxins, and potential applications of snake species found in Colombia. Data compilation was conducted according to the PRISMA guidelines, and the systematic literature search was carried out in Pubmed/MEDLINE. Venom proteomes from nine Viperidae and three Elapidae species have been described using quantitative analytical strategies. In addition, venoms of three Colubridae species have been studied. Bioactivities reported for some of the venoms or isolated components—such as antibacterial, cytotoxicity on tumoral cell lines, and antiplasmodial properties—may be of interest to develop potential applications. Overall, this review indicates that, despite recent progress in the characterization of venoms from several Colombian snakes, it is necessary to perform further studies on the many species whose venoms remain essentially unexplored, especially those of the poorly known genus *Micrurus*.

## 1. Introduction

Snake venom is a complex biological secretion of specialized glands in certain snake species used to subdue prey or as a defense mechanism [1]. Venoms mainly comprise enzymes, proteins without enzymatic activity, peptides, organic chemical compounds, trace molecules such as ions (iron, calcium, zinc, cobalt, potassium or sodium), and citrate, among other components [1]. Snakebite envenoming occurs when snakes accidentally inject venom into humans, causing various pathophysiological effects, some of which can be life-threatening [2,3,4]. Nevertheless, snake venoms are also rich sources of bioactive compounds, which have led to the development of some therapeutic drugs [1,5]. 

Among more than 4000 snake species reported worldwide, around 20% are venomous [6,7]. Colombia is a megadiverse country, with over 270 snake species recognized [8]. Nevertheless, snakes from the Viperidae and Elapidae families and a reduced number of the Colubridae can inject venom; therefore, they are considered of medical importance [9,10]. In Colombia, between 4000 and 5500 snakebites have been reported annually in recent years [10], most of them inflicted by *Bothrops* spp.; however, the most severe cases are attributed to snakes from the genus *Micrurus* spp., *Lachesis* spp., and Colombian rattlesnake *Crotalus durissus cumanensis* [9,10]. Studies about diversity and abundance in snake venom proteomes report two dominant protein families for elapid venom: three-finger toxins (3FTxs) and phospholipases A_2_ (PLA_2_s). In contrast, the most abundant protein families for the Viperidae family are PLA_2_s, snake venom serine proteases (SVSPs), and snake venom metalloproteases (SVMPs) [11,12]. 

Snake venom composition can vary significantly between species and even among individuals within a species, age, and geographical location, among other aspects [1,13]. Therefore, it is crucial to characterize the venom composition for species of a country or a geographical area within a country. In addition, it is essential to characterize the snake venoms functionally, to correlate the signs and symptoms observed in envenomed patients, and to contribute to improving antivenom production [3]. The first study about snake venoms from Colombia was conducted in the beginning of the 1990s [14]. In more recent years, the composition of a number of these venoms have been characterized by proteomic profiling studies, and the isolation and characterization of some of their toxins have been achieved [14]. 

This systematic review aims to summarize and discuss the available information on biological effects, proteomic composition, and potential use of snake venoms and toxins from Colombia.

## 2. Materials and Methods

This review was conducted and reported according to the PRISMA (Preferred Reporting Items for Systematic Review and Met-Analysis) guidelines.

### 2.1. Eligibility Criteria, Search Strategies, and Information Sources

Published articles about whole snake venom characterization, composition by proteomic strategies, isolation and characterization of snake toxins, and bioprospecting uses were included. As exclusion criteria, we eliminated reviews, conference abstracts, editorials, and articles on antivenoms, natural or synthetic snake venom inhibitors, clinical aspects of envenomings, other toxinology topics, publications not related to snake venoms or toxins from Colombia, and articles without full-text access. 

A systematic search using the described strategies was carried out in Pubmed/MEDLINE. Searches were limited without a start date through 31 July 2023, to identify published articles about whole snake venom characterization, composition by proteomic strategies, isolation and characterization of toxins, and their bioprospecting uses. The following search terms were used: (“Toxins, Biological”[Mesh] AND “Colombia”) OR (“Snake Venoms”[Mesh] AND “Proteomics”[Mesh] AND “Colombia”). This review was not registered in the final version.

### 2.2. Study Selection

Search strategies were imported and merged into reference management software, ZOTERO^®^. Then, titles and abstracts were assessed for inclusion. Full texts of relevant articles were retrieved and independently assessed by two authors.

### 2.3. Data Collection Process and Data Items

Data were extracted independently by two of the authors and compared using standardized data extraction forms. Discrepancies were discussed. Data extracted included title, authors, journal, and relevant comments.

### 2.4. Graphics

Graphics were built with Python (Version 3.10).

## 3. Results and Discussion

From the systematic literature search, 512 manuscripts were retrieved. However, after reviewing all articles, 467 were excluded because they either were not related to snake venoms and toxins from Colombia (*n* = 399), dealt with clinical aspects of snakebites (*n* = 8), or were associated with other topics in toxinology (antivenoms, *n* = 2; *Lonomia* identification, *n* = 14; scorpion venoms, *n* = 17; snake venom/toxin inhibitors, *n* = 17 and spider venoms, *n* = 9). In addition, it was impossible to access the full text of one manuscript. A total of 45 articles were chosen, and from their list of references, 7 were included (Figure 1). 

### 3.1. Proteomic Profiles of Snake Venoms from Colombia

Proteomic techniques have been used to unveil the venom compositions, and to estimate the relative abundances of protein families for the Viperidae species shown in Figure 2. The venoms of *B. asper* [15,16], *B. atrox* [17], *B. punctatus* [18], *B. ayerbei* [16], *B. rhombeatus* [15], *Bothrocophias myersi* [19], *Porthidium lansbergii* [20], *Lachesis acrochorda* [21], and *C. d. cumanensis* [22] have been characterized by proteomic profiling. Proteins identified in these venoms belong to between seven to twelve families, with a median of 9. In general, the most abundant components are PLA_2_s, SVMPs, SVSPs, and L-amino acid oxidases (LAAOs). However, there were exceptions. For example, the venom of *C. d. cumanensis* has a high abundance of crotoxin (64.71%), in similarity with other South American rattlesnakes [23,24].

All *Bothrops* spp. venoms characterized until now have higher proportions of SVMPs than PLA_2_s [15,16,17,18]. This same pattern was observed in the venoms of *P. lansbergii* and *L. acrochorda* [20,21]. In contrast, the venom of *B. myersi* has a superior percentage of PLA_2_s. In addition, the venom of this species is the most complex, with twelve protein families [19]. Special characteristics of *L. acrochorda* venom are their high amounts of SVSPs and bradykinin-potentiating peptides (BBPs, 35.1% and 25.5%, respectively), even higher than PLA_2_s. Moreover, this venom showed the highest abundance of LAAOs (9.6%) [21]. 

C-type lectins are more abundant in the venom of *B. punctatus* (16.7%) [18], and they were identified in all Viperidae venoms. Conversely, the venom of *P. lansbergii* has the highest amounts of disintegrins [20], whereas these toxins were not identified in the venom of *L. acrochorda* [21]. A particular finding is the presence of Hyaluronidases, the spreading factor of venoms, which were only detected in the venom of *B. myersi* at a trace proportion (0.01%) [19]. However, this result might be explained because, in some venoms, not all proteins were successfully identified or assigned to known protein families. 

The venom of *C. d. cumanensis* differs from all others studied thus far because this species is the only viper in Colombia that causes neurotoxicity. This effect is explained by the presence of high amounts crotoxin (64.7%) in the venom, which induces diaphragm-flaccid paralysis in snakebites inflicted by this Colombian rattlesnake [9]. In similarity to venoms from several other rattlesnakes, *C. d. cumanensis* venom also contains crotamine, a small basic polypeptide with myotoxic and cell-penetrating activities [25]. 

In addition to those shown in Figure 2, other proteomic studies have been performed using a “bottom-up whole venom shotgun profiling” approach, which can identify the proteins present in the venoms, but does not generate a reliable quantification of their relative abundance discussed in [26]. Nevertheless, the qualitative information provided by such analyses is valuable. Jiménez-Charris et al. [27] characterized *B. asper* venom from the Gorgona island and two ecoregions from Valle del Cauca state (Pacific and western). In the venom from Gorgona island, the following protein families were identified: SVMPs, PLA_2_s, CTL, CRISP, LAAO, SVSP, PLB, NGF, and Glutaminyl cyclase (GCY). In terms of qualitative composition, the venom of Gorgona Island is comparable to that of the Pacific ecoregion. The main difference was the presence of Nuc in the latter snake venom. In contrast, in the venom from the western ecoregion, no PLB, GCY, and Nuc were identified. Despite these differences, the three venoms studied are comparable with *B. asper* venoms from other regions of Colombia [15] in terms of protein family presence. 

Another study by Montoya-Gómez et al. [28] described the qualitative protein family composition of *L. acrochorda* venom from Valle del Cauca, and detected SVMPs, PLA_2_s, SVSPs, LAAO, CTL, and NGF. This venom was very different from that reported by Madrigal et al. [21], which contained SVMPs, PLA_2s_, SVSPs, LAAO, Nuc, Hya, PDE, CTL, CRISP, BPP, and VEGF. The variations between the two venoms were attributed to the different proteomic approaches used, as well as the source of venoms from specimens kept in captivity and from wild animals for the studies of Madrigal et al. [21] and Montoya-Gómez et al. [28], respectively. 

On the other hand, the composition of three venoms from the Elapidae family (coral snakes) have also been studied by proteomic profiling (Figure 3). 

Proteomic analyses of *Micrurus* spp. venoms have revealed a dichotomic compositional pattern, with some species containing more PLA_2s_ than 3FTxs, while in others, 3FTx predominate [32]. *Micrurus* species inhabiting South America tend to express the 3FTx-predominant venom phenotype, while the PLA_2_-rich pattern is observed in species inhabiting North America. In contrast, species found in Central America and northern South America present either of the two venom patterns [32]. *Micrurus* spp. from Colombia conform to the venom compositional pattern described. *M. mipartitus* from Antioquia state (northwestern Colombia) is rich in 3FTxs (61.1%), but it also contains 29.0% of PLA_2_s [29]. Instead, the venom of *M. dumerilii* from Antioquia is predominant in PLA_2_s (52.0%), compared to 28.1% of 3FTxs [30]. Similar findings were reported for the *M. lemniscatus helleri* venom from Amazonas (the southernmost state from Colombia), with 62.5% of PLA_2_s and 21.1% of 3FTxs [31]. The other components identified in *Micrurus* spp. venoms from Colombia were always below 10.0%.

### 3.2. Biological Activities of Viperidae and Elapidae Venoms from Colombia

The lethal activity of Viperidae venoms has been determined in mouse assays. The most studied venom is *B. asper* from different geographical regions in the country. The LD_50_ (95% confidence interval, µg/mouse) values by intraperitoneal (i.p.) route, for *B. asper* from Antioquia, Cauca, Valle del Cauca (Pacific ecoregion), and Gorgona Island are 67.1 (60.1–74.1) [33], 100.9 (83.2–122.8) [16], 112.8 (96.6–129.6) [27], and 118.2 (86.5–158.0) [27], respectively. It is also important to consider that the average of the LD_50_ (µg/mouse) of *B. asper* venoms from different ecoregions from Antioquia state is 65.3 (51.0–89.0) [3]. On the other hand, the LD_50_ (µg/mouse) values for *B. atrox* (Meta); *B. ayerbei* (Cauca); *B. punctatus* (Antioquia); and *B. rhombeatus* (Cauca), are 81.4 (80.2–83.6) [33], 50.1 (37.4–58.3) [16], 47.0 (36.0–61.0) [34] and 54.9 (36.0–83.8) µg/mouse [35], respectively. Therefore, the most lethal venoms from *Bothrops* spp. tested until now are *B. ayerbei*, *B. rhombeatus*, and *B. punctatus*. However, these results are not completely comparable because the biological assays use different mice strains. 

The LD_50_ (95% confidence interval, µg/mouse) values by i.p. route of the other Viperidae venoms are: *L. acrochorda* (Valle del Cauca) 290.0 (260.0–323.0) [28]; *L. acrochorda* (Chocó) 130.0 (106.0–160.0) [34]; *P. nasutum* (Antioquia) 62.0 (51.0–74.0) [35]; *P. lansbergii* (Atlántico), yellow and gray morphs 98.0 (87.7–109.5) and 93.9 (730–128.6), respectively [36]; *B. myersi* (Valle del Cauca) 128.4 (101.1–156.6) [19]; and *C. d. cumanensis* (pool Colombia) 1.8 (1.2–2.5) µg/mouse [34]. 

In contrast, the LD_50_ (95% confidence interval, µg/mouse) for the venom of *M. dumerilii* (Antoquia) is 23.6 (15.0–38.0) [30], and for *M. mipartitus* (Antioquia) is 9.0 (6.6–12.1) [34].

Several studies have characterized other biological activities of Viperidae venoms from Colombia (Figure 4). 

Edema-forming activity (in the mouse footpad assay) has been demonstrated for all venoms, and there is a good correlation with the effects observed in the snakebites inflicted for the species described in Figure 4. Edema can range from mild to severe. A mild effect is induced by *B. schelegelii* [37] and *C. d. cumanensis* [34,38] venoms, while a moderate effect is caused by *B. myersi* venom [19], and a severe effect is triggered by *Bothrops* spp., *Porthidum* spp., and *Lachesis* spp. venoms [9,16,20,34,39,40,41].

Myotoxicity has been evaluated by quantifying the activity of creatine kinase (CK) in the plasma of mice injected with venoms or toxins. All Viperidae venoms have demonstrated myotoxicity in mouse assays [16,19,34,36,41], in agreement with the clinical finding of moderate to severe CK activity increase in the plasma of envenomed patients [9,19,38,39,40]. 

The hemorrhagic activity of snake venoms is attributed to the action of SVMPs [42]. All Viperidae venoms tested until now have hemorrhagic activity; however, their potency is variable. *Bothrops* spp. [16,27,34], *Porthidium* spp. [20,34], *L. acrochorda* [28,34], and *B. myersi* venoms are more hemorrhagic than *B. schelegelii* venom [34,37]. In contrast, *C. d. cumanensis* is only weakly hemorrhagic [34]. In addition, Viperidae snake venoms from Colombia induce hemostatic disorders related to their pro-coagulant activity demonstrated in vitro, their fibrinogenolytic activity displayed in vivo, and the alteration of coagulation times [16,19,20,27,34,36]. These venoms are characterized by a massive consumption of fibrinogen that can only be recovered by antivenom injection [9,39,43,44].

Venoms from *Micrurus* spp. are mainly neurotoxic; however, they have been tested for their capacity to induce other activities. The venom of *M. dumerilii* induced a conspicuous myotoxic, cytotoxic, and anticoagulant effect, and it was mildly edematogenic and proteolytic, whereas it lacked hemorrhagic activity [30]. In contrast, *M. mipartitus* venom caused weak anticoagulant and myotoxic effects and lacked hemorrhagic activity [29]. The activities described correlate with signs and symptoms observed in snakebites inflicted by coral snakes because they do not induce local tissue effects, but they cause systemic neurotoxicity, leading to a life-threatening flaccid paralysis of the diaphragm [9,45].

### 3.3. Comments on Specific Genus/Species

Several studies have been performed to isolate and characterize toxins and, furthermore, try to correlate the results with the effects observed in patients who suffer a snakebite. In addition, other studies have been carried out to identify venom variations at different levels.

#### 3.3.1. *Bothrops* spp.

As mentioned above, venom from *B. asper* is the most studied. In this context, different toxins from *Bothrops* spp. have been isolated. Posada Arias et al. [46] isolated and characterized an acidic PLA_2_ from the *B. asper* venom (Antioquia). The toxin was named BaCol PLA_2_. This protein induced apoptosis, indirect hemolysis, and anticoagulant activity in vitro, and produced edema and myotoxicity in mice. Another PLA_2_ was identified by Pereañez et al. [47]. This basic enzyme induced a conspicuous myotoxic effect and a moderate edema. In vitro, the toxin was cytotoxic and weakly anticoagulant. The biological effects described for these PLA_2_s contribute to the signs and symptoms observed in patients bitten by *B. asper* [9,39].

Other studies characterized proteins from *B. atrox* venom (Meta). Núñez et al. [48] isolated a PLA_2_ homolog (a PLA_2_ without enzyme activity, due to amino acid changes at the active site and Ca^2+^-binding loop [49]). This toxin lacked anticoagulant activity; nevertheless, it induced myotoxicity and edema. From *B. atrox* venom (Meta), Patiño et al. [50] characterized a hemorrhagic SVMP, which also caused myotoxicity and degraded fibrinogen. The two toxins may play a role in myotoxic, hemorrhage, and clotting disorders observed in patients who suffer a snakebite perpetrated by *B. atrox* [9]. 

Saldarriaga et al. [33] studied the ontogenic variation in *B. atrox* (Meta) and *B. asper* (Antioquia) venoms. The study analyzed venoms of <0.5, 1, 2, and 3 years of both species. A conspicuous ontogenetic variability was observed in venoms from both species. Venoms from newborn and juvenile specimens showed higher lethal, hemorrhagic, edema-forming, and coagulant activities, whereas 3-year-old specimens showed higher PLA_2_ activity. Other differences were evidenced in the molecular masses of proteins expressed. A predominance of proteins with high molecular mass was observed in the venoms from specimens of <1 year of age, with a change towards proteins having lower molecular mass as snakes aged. 

#### 3.3.2. *Lachesis acrochorda*

Ángel-Camilo et al. [51] reported the cardiovascular effects induced by *L. acrochorda* venom (Cauca). In vitro, the venom was not cytotoxic to neutrophils and platelets but triggered human plasma coagulation and platelet aggregation. Ex vivo, venom increased the magnitude of spontaneous contractions of the isolated right atrium of rats. In contrast, venom relaxed KCl- or phenylephrine-induced contractions in isolated rat aorta. In addition, venom caused hypotension and bradycardia in rats. It also detected hemorrhage in pulmonary and renal tissues. The authors suggested that the presence of SVMPs and SVSPs in the venom may explain these cardiovascular effects. Nevertheless, the hypotensive action of this venom can also be attributed to the abundant BPP’s (21.5%) [21], the highest proportion reported in proteomic studies performed on Colombian Viperidae venoms (Figure 2).

Otero et al. [41] compared the lethal, hemorrhagic, edema-forming, myotoxic, coagulant, defibrinating, proteolytic, and indirect hemolytic activities of *Lachesis* spp. from Costa Rica, Colombia, and Brazil. At that moment, *L. acrochorda* from Antioquia and Chocó was classified as *L. muta muta*. All venoms tested showed the biological activities mentioned above. Even though significant differences were observed in specific pharmacological activities between some of the venoms, the authors concluded that there was no consistent pattern of variation suggesting a divergence of one venom from the others.

#### 3.3.3. *Crotalus durissus cumanensis*

The venom of this species has been studied to evaluate ontogenic and geographical variations, and some of its toxins have been isolated. Cespedes et al. [52] compared the venoms of the mother (Tolima), father (Guajira), and offspring (six specimens) in their capacity to induce lethal, edema-forming, defibrinating, hemolytic and coagulant activities. All venoms induced edema, but none produced a hemorrhagic effect. Venom of the mother was more lethal, hemolytic, coagulant, and defibrinating than the father’s venom. In contrast, venoms from young snakes were comparable to that obtained from the mother, but the coagulant effect was stronger in offspring venoms. Electrophoretic profiles of all the venoms were not significantly different. However, chromatographic profiles revealed differences for the father’s venom, while the mother’s and offspring venoms were similar. The authors concluded that the venom variability in *C. d. cumanensis* did not appear to be associated with age and gender. On the other hand, Arévalo-Páez et al. [53] found that venoms from both adult and juvenile snakes showed neurotoxic activity on chick biventer cervicis nerve-muscles preparations, but this effect developed more rapidly with juvenile than adult venoms. 

Regarding geographical variability, Rodriguez-Vargas et al. [54] identified differences between *C. d. cumanensis* from different eco-regions of Colombia. The main finding was the presence of crotamine only in venoms from the Caribbean region. In contrast, the venom of Magdalena Medio was the most lethal and coagulant, and it showed the highest PLA_2_ and hyaluronidase activities. 

The neurotoxicity of *C. durissus* sub-species is attributed to the presence of the crotoxin complex, a heterodimer, with an acidic sub-unit (crotapotin) and a basic PLA_2_ named CB [55,56]. Pereañez et al. [57] isolated the CB sub-unit from *C. d. cumanensis* (Meta) venom. The N-terminal sequence of this protein showed high identity with CBs from other South American rattlesnakes. In addition, the enzyme induced a conspicuous myotoxicity and moderate edema, and it caused human plasma anticoagulation. Two SVSPs (Cdc SI, and Cdc SII) have also been isolated from *C. d. cumanensis* (Meta) venom [58]. The N-terminal sequences of the two toxins suggested that they belong to the family of thrombin-like enzymes. These toxins showed coagulant activity on human plasma and fibrinogen, moderate edema induction, and increased vascular permeability and defibrinogenation. Nevertheless, they lack hemorrhagic and myotoxic activities. From the same venom, a LAAO was purified [59]. This enzyme lacked cytotoxic activity on mouse myoblasts (C2C12) and peripheral blood mononuclear cells but showed antibacterial activity (see below). Studies on isolated *C. d. cumanensis* venom components indicate that CB from the crotoxin complex contributes to systemic myotoxicity and edema observed in patients envenomed by this species, and the two SVSPs contribute to clotting disorders [9]. 

#### 3.3.4. *Bothriechis schelegelii*

Otero et al. [34] and Prezotto-Neto et al. [37] reported the biological activities of *B. schlegelii* venom from Antioquia (Figure 4). In addition, Montealegre-Sánchez et al. [60] reported some individual variability in this species by studying the venoms of two females and one male collected in Valle del Cauca State, the southwest region of Colombia. Venoms showed differences in electrophoretic and chromatographic profiles. The venoms displayed indirect hemolytic, edematogenic, and procoagulant activities, with differences between the male and females. Further, none of the venoms caused hemorrhage at the tested doses (20–80 µg), which differs from the findings of Otero et al. [34] and Prezotto-Neto et al. [37]. 

A LAAO was purified from *B. schelegelii* venom (Antioquia) [61]. This enzyme lacked cytotoxic activity on mouse myoblasts (C2C12) and peripheral blood mononuclear cells. In addition, this protein showed antibacterial activity (see below). 

#### 3.3.5. *Porthidium* spp.

Jiménez-Charris et al. [62] tested the systemic alterations triggered by *P. lansbergii* venom (Caribbean region). After intraperitoneal injection, envenomed mice showed hypodynamic condition, clonic head movements, bradypnea, and thoracoabdominal imbalance. Histological examinations showed that the venom caused brain and lung hemorrhage, and the liver evidenced parenchymal alterations with abundant extravasated erythrocytes. Kidneys showed tubular necrosis; furthermore, increased plasma creatinine was observed. After 12 h, envenomed mice showed increased alkaline phosphatase and alanine aminotransferase enzymatic activities. In contrast, aspartate aminotransferase and lactate dehydrogenase increased at seven h and returned to near baseline by 12 h. These results suggest that *P. lansbergii* induces systemic hemorrhage that can trigger hypovolemic shock, which, together with kidney injury, can contribute to acute renal injury observed in patients who suffer *P. lansbergii* snakebites [9]. 

From the same venom, two PLA_2_ enzymes were characterized, one basic (Pllans-I) and another acidic (Pllans-II) [63]. The basic PLA_2_ caused myotoxicity, edema, and lethality (by intracerebroventricular injection) in mice; in vitro, it showed cytotoxic and anticoagulant activities. In contrast, the acidic enzyme lacked all these activities, except for the induction of moderate edema. The authors also reported a synergism between two PLA_2_s for the myotoxic effect. Another acidic PLA_2_ (PnPLA2) was purified by Vargas et al. [64] from the venom of *P. nasutum* (Antioquia). This enzyme was not cytotoxic on murine skeletal muscle myoblast C2C12. Nonetheless, it inhibited platelet aggregation. In addition, PnPLA_2_ showed antibacterial activity (see below).

#### 3.3.6. *Micrurus* spp.

Renjifo et al. [65] tested the neurotoxic activity of *M. mipartitus* and *M. dissoleucus* venoms on chick biventer cervicis nerve-muscle preparations. The venom of *M. mipartitus* induced inhibition of nerve-mediated twitches and blocked the contractile response to exogenous acetylcholine (Ach), which indicated a postsynaptic mode of action. In contrast, *M. dissoleucus* venom did not cause complete inhibition of nerve-mediated twitches and inhibited the contractile response to exogenous Ach. In addition, both venoms showed myotoxic activity on chick biventer cervicis nerve-muscle preparations, confirmed by histological examination, with vacuolization, edema, and necrotic cell infiltration. 

The most studied venoms of *Micrurus* spp. From Colombia are those of *M. mipartitus* and *M. dumerilii*, and some components that contribute to their toxic effects have been isoalated. In *M. mipartitus* venom, the most abundant toxin is a 3FTx named Mipartoxin-I, which showed a potent lethal effect in mice and blocked the postsynaptic nicotinic receptor on both avian and mouse nerve-muscle preparations [66], in agreement with findings of Renjifo et al. [65]. Another lethal protein is MmipPLA_2_, which is also myotoxic in mice [67]. Recently, it was reported that the lethal effect of the whole venom was completely neutralized when a mixture of antibodies raised against the mentioned toxins was used [68]. From the venom of *M. mipartitus*, a LAAO has also been isolated, named MmipLAAO [69], which is not likely to be related to the lethal effect of the whole venom, but it showed antimicrobial activity (see below). 

From the venom of *M. dumerilii*, a PLA_2_ was isolated and named MdumPLA_2_, which was not lethal but strongly myotoxic and moderately edematogenic [67]. This toxin has been cloned and expressed in *Escherichia coli* [70]. The recombinant enzyme showed catalytic, anticoagulant, edematogenic, and myotoxic activities. In addition, it was used as an immunogen to produce antibodies in rabbits, which neutralized the PLA_2_ activity of the recombinant toxin and a moderate percentage of the myotoxic activity of *M. dumerilii* whole venom. The authors proposed that including recombinant proteins in the immunizing mixtures may be a strategy to improve antivenom production against *Micrurus* spp. venom. 

*M. dumerilii* venom also contains a 3FTx named Clarkitoxin-I-Mdum, which is not lethal [71]. More recently, a lethal fraction of this venom was partially purified, which contains two 3FTx and one PLA_2_. It was demonstrated that antibodies produced against this fraction neutralized the lethal effect induced by *M. dumerilii* whole venom [72]. 

### 3.4. Colubrid Venoms

The venoms of the Colubridae family have been underexplored in Colombia. Only three species have been studied: *Erythrolamprus bizona*, *Pseudoboa neuwiedii*, and *Leptodeira annulata*.

*L. annulata* venom induced a partial neuromuscular blockade in chick biventer cervicis neuromuscular preparations in vitro, accompanied by morphological alterations, presumably attributable to the pronounced proteolytic activity mediated by SVMPs [73], as recorded on substrates such as against elastin-Congo red, fibrin, fibrinogen, gelatin, and blue skin powder. In contrast, the venom did not show esterase activity towards the substrate BapNA, indicating the absence of SVSPs, and consistent with the absence of thrombin-like activity (no coagulation in citrated plasma or purified fibrinogen). Likewise, this venom did not induce platelet aggregation and LAAO activity. Furthermore, the venom elicited myonecrosis and elevated serum CK concentrations, and its PLA_2_ activity was confirmed through attenuation in the presence of a specific PLA_2_ inhibitor [74].

On the other hand, the venom of *P. neuwiedii* induced adverse effects on neurotransmission in chick biventer cervicis neuromuscular preparations in vitro. It produced a moderate blockade and reduced muscle contractures when exposed to exogenously added acetylcholine and potassium chloride. Additionally, it caused mild muscle damage [75]. This venom contains primarily SVMPs, CRISPs, and PLA_2_ enzymes, as well as less abundant components such as C-type lectin-like protein (CLP), phospholipase B (PLB), and vascular endothelial growth factor (VEGF). Notably, no serine proteinases (SVSPs) were found in the venom. From an enzymatic perspective, the venom exhibits high proteolytic activity on substrates like casein, azocasein, and gelatin. This proteolytic activity can potentially affect coagulation in vivo by degrading fibrinogen through the action of SVMPs. The PLA_2_ activity in the venom was comparable to that of *B. atrox* venom [76].

The venom of *E. bizona* exhibited high proteolytic activity compared to the venom of *P. neuwiedii*, with very low PLA_2_ and amidolytic activities. Additionally, this venom provoked a partial neuromuscular blockade that was not accompanied by alterations in twitch height. This suggests that the blockade likely resulted from myotoxicity rather than a neurotoxic origin, although there could be a masked post-sympathetic effect [75].

In conclusion, these venoms exhibited diverse enzymatic and biological activities, with local effects primarily mediated by SVMPs and PLA_2_ enzymes, and none showed activities related to serine proteinases.

### 3.5. Potential Applications of Colombian Snake Venoms

Snake venoms have a recognized therapeutic potential and have been extensively studied for developing new drugs. Captopril (antihypertensive), tirofiban and Eptifibatide (antiplatelet), and Batroxobin (thrombolytic) are examples of approved drugs derived from snake venoms [1]. Screenings on Colombian snake venoms have demonstrated biological activities such as antibacterial, antiplasmodial, and cytotoxic properties on tumoral cell lines. These studies will be summarized below.

#### 3.5.1. Antibacterial Activity

Vargas et al. [64] reported the isolation of an acidic PLA_2_ (PnPLA_2_) from *P. nasutum* venom with a dose-dependent bactericidal activity against *Staphylococcus aureus*. The minimum inhibitory concentration (MIC) and Minimal Bactericidal Concentration (MBC) were 32 μg/mL. This PLA_2_ was not cytotoxic to murine skeletal muscle myoblasts C2C12, in contrast to basic enzymes isolated from other viperid snake venoms, suggesting its potential pharmacological applications [64]. 

Further work by Vargas et al. [59] reported an L-amino acid oxidase (CdcLAAO) from *C. d. cumanensis* venom with antibacterial activity against *S. aureus* (Gram-positive) and *Acinetobacter baumannii* (Gram-negative) bacteria, with MICs of 8 μg/mL and 16 μg/mL for *S. aureus* and *A. baumannii*, respectively. Scanning electron microscopy revealed morphological alterations in bacterial cells treated with CdcLAAO for 24 h, consistent with membrane damage and debris deposition on the cell surface. Interestingly, CdcLAAO did not exhibit cytotoxic activity on the mouse myoblast cell line C2C12 and peripheral blood mononuclear cells.

Later, Vargas et al. [61] described the antibacterial activity of BsLAAO, a LAAO isolated from *B. schlegelii* venom. This toxin showed an inhibitory effect against *S. aureus* with a MIC of 4 μg/mL and MBC of 8 μg/mL. Against *A. baumannii*, it showed a MIC of 2 μg/mL and MBC of 4 μg/mL. This activity was inhibited by catalase, indicating that antimicrobial activity was due to H_2_O_2_ production. BsLAAO did not show cytotoxic activity against mouse myoblast cell line C2C12 or peripheral blood mononuclear cells. The existence of a window of concentrations in which LAAOs exert bactericidal action but are harmless to human cells demonstrates the potential antimicrobial applications of these toxins. To establish their inhibitory mechanism and eventual therapeutic uses, it is necessary to carry out further studies.

Finally, the antibacterial effect of a purified LAAO (MipLAAO) from *M. mipartitus* venom was demonstrated. It showed a potent bactericidal effect on *S. aureus* (MIC: 2 µg/mL), but not on *E. coli* [69]. 

#### 3.5.2. Antiplasmodial Activity

Two studies have reported the antiplasmodial activity of snake venoms from Colombia and their isolated toxins. Quintana et al. [77] showed the antiplasmodial activity of the whole venom of *C. d. cumanensis*, a fraction containing crotoxin, and purified crotoxin B against *Plasmodium falciparum* in vitro. The whole venom was active against the parasite at concentrations of 0.17 ± 0.03 μg/mL, crotoxin complex fraction at 0.76 ± 0.17 μg/mL, and Crotoxin B at 0.6 ± 0.04 μg/mL. Crude venom and crotoxin of *C. d. cumanensis* are strongly neurotoxic. However, the PLA_2_ subunit of crotoxin, Crotoxin B, shows negligible neurotoxicity, even at doses as high as 700 mg/kg in mice, suggesting the potential antiplasmodial activity of this PLA_2_.

Further work by Quintana et al. [78] studied the antiplasmodial effect of the whole venom and two fractions purified by ion-exchange chromatography from *Bothrops asper* venom (fraction V contained a catalytically active PLA_2_, and fraction VI contained a PLA_2_ homolog devoid of enzymatic activity). The whole venom, as well as its fractions V and VI, were active against cultures of *P. falciparum* at concentrations of 0.13 ± 0.01 μg/mL, 1.42 ± 0.56 μg/mL, and 22.89 ± 1.22 μg/mL, respectively. Assays of cytotoxic activity on peripheral blood mononuclear cells found that fraction V had higher toxicity than whole venom and fraction VI, the latter showing a more selective antiplasmodial potential [78]. 

#### 3.5.3. Cytotoxicity on Tumoral Cell Lines

Several studies have established the antitumoral activity of different toxins isolated from Colombian venoms, mainly from the *Porthidium* genera. The first study was reported by Bonilla-Porras et al. [79]. A zinc-metalloproteinase (Nasulysin-1) purified from *P. nasutum* venom showed specific apoptosis-inducing activity in acute lymphocytic leukemia and chronic myeloid leukemia cells, without affecting the viability of human lymphocyte cells. In addition, Nasulysin-1 at a concentration of 20 μg/mL induced loss of the mitochondrial membrane potential, activated the apoptosis-inducing factor, the protease caspase-3, and induced chromatin condensation and DNA fragmentation, all markers of apoptosis. These results suggested the potential of this metalloproteinase as a therapeutic agent for treating leukemia [80].

On the other hand, Jiménez-Charris et al. [80] reported the antitumoral and angiostatic potential effects of an acidic Asp49–PLA_2_ (Pllans–II) from *P. langbergii* snake venom on HeLa cells in vitro. This toxin exhibited dose-dependent cytotoxicity and cell cycle arrest in the G1 phase on cervical carcinoma HeLa cells without effects on normal epithelial and endothelial cells. Pllans–II induced both early and late apoptosis on HeLa cells through the modulation of essential gene mediators of apoptosis through extrinsic pathways [80]. Later, Montoya-Gómez et al. reported that Pllans–II induced cell death in a cervical cancer cell line. This toxin showed a dose-dependent cytotoxic effect on cancer cells and an insignificant effect on normal endothelial cells [81]. 

A disintegrin with antitumoral activity has also been Isolated from *P. lansbergii* venom. This toxin, named Lansbermin-I, has the RGD motif in its sequence. Lansbermin-I showed potent inhibition of ADP and collagen-induced platelet aggregation on human plasma and displayed inhibitory effects on the adhesion and migration of breast cancer cell lines without affecting nontumorigenic breast and lung cells. Additionally, Lansbermin-I inhibited in vitro angiogenesis on human endothelial (HUVEC) cells [82]. These reports are very promising for developing an antitumoral agent derived from *P. lansbergii* venom.

Finally, the cytotoxic effect of *M. mipartitus* snake venom and a purified LAAO (MipLAAO) on human peripheral blood lymphocytes and Jurkat cells was reported. *M. mipartitus* venom and MipLAAO induced morphological changes in the cell nucleus/DNA, mitochondrial membrane potential, intracellular reactive oxygen species levels, and cellular apoptosis markers in a dose-dependent manner, without affecting human peripheral blood lymphocytes [83]. This is the only report that shows a potential application for *Micrurus* venom from Colombia. 

#### 3.5.4. Distribution of the Species Described in This Systematic Review

Figure 5 shows the geographical distribution of the venoms of the Viperidae and Elapidae families described in this systematic review. 

#### 3.5.5. Reported Findings after the Final Date of the Systematic Search

During the reviewing process of this manuscript, three venom proteomes of coral snakes were reported [84]. The studied venoms were *M. helleri* (Putumayo), *M. medemi* (Meta), and *M. sangilensis* (Santander). All venoms were rich in PLA_2_s, with relative contents of 40.63%, 43.14%, and 30.40% for *M. helleri*, *M. medemi*, and *M. sangilensis*, respectively. The second family of proteins was the 3FTXs in all venoms, with percentages between 14.10 and 17.69%. An intriguing finding was the content of SVMPs, which were between 9.63% and 13.10%. Nevertheless, the proteolytic activity of the venoms was lower than trypsin, and the hemorrhagic activity was not tested. Therefore, these results need further studies. The other venom components, such as L-amino acid oxidase, Phospholipase B-like, and serine protease, were below 10%.

## 4. Conclusions

Research on Colombian snake venoms has thus far reported the quantitative proteomic composition of nine Viperidae species (*B. atrox*, *B. asper*, *B. ayerbei*, *B. rhombeatus*, *B. punctatus*, *P. lansbergii*, *B. myersi*, *C. d. cumanensis*, and *L. acrochorda*). The most abundant toxins in those venoms are PLA_2_s and SVPMs, with exceptions for *L. acrochorda*, in which the most abundant toxins are SVSPs. These results correlate with the pathophysiological effects observed in Viperidae envenomings in Colombia, including edema, hemorrhage, hemostatic disorders, and myotoxicity. Furthermore, neurotoxicity is the main effect observed in crotalic envenomings due to the high abundance of neurotoxic PLA_2_ (crotoxin).

On the other hand, three venoms from Colombian coral snakes have been studied at the compositional level. The venom of *M. mipartitus* showed 3FTxs as the main component. In contrast, the venom of *M. dumerilii* and *M. lemniscatus helleri* demonstrated a pattern dominated by PLA_2_s. The main effect reported for *Micrurus* venoms is neurotoxicity, an effect that correlates with the biological activities of their main toxins. 

The venoms of only three Colubridae species have been studied: *Erythrolamprus bizona*, *Pseudoboa neuwiedii*, and *Leptodeira annulata*. These showed diverse enzymatic and biological activities, with local effects mediated by SVMPs and PLA_2_ enzymes. 

Although 15 venoms of snake species from Colombia have been characterized, there is a need to perform further studies on the many uncharacterized species, especially those from the *Micrurus* genus. In addition, it is important to further explore the potential applications of Colombian snake venoms. To this date, antibacterial, antitumoral, and antiplasmodial activities have been reported. It is crucial to expand research to evaluate the possible therapeutic applications of Colombian snake venom towards developing novel pharmaceutical drugs.

Finally, the information summarized in this review can be used by toxinologists, biologists, herpetologists, and physicians in Colombia, and in other countries where the species described in this study also inhabit, with an interest in understanding the venoms’ composition, their biological activities, and the pathophysiology of snakebite envenomings. 

## Figures and Tables

**Figure 1 toxins-15-00658-f001:**
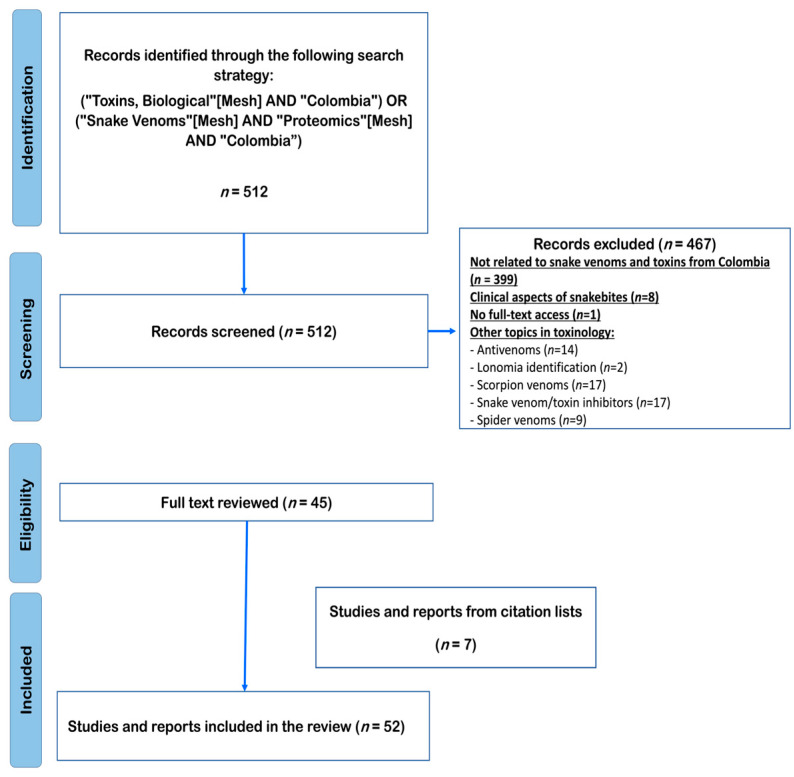
PRISMA flow diagram for the literature search strategy.

**Figure 2 toxins-15-00658-f002:**
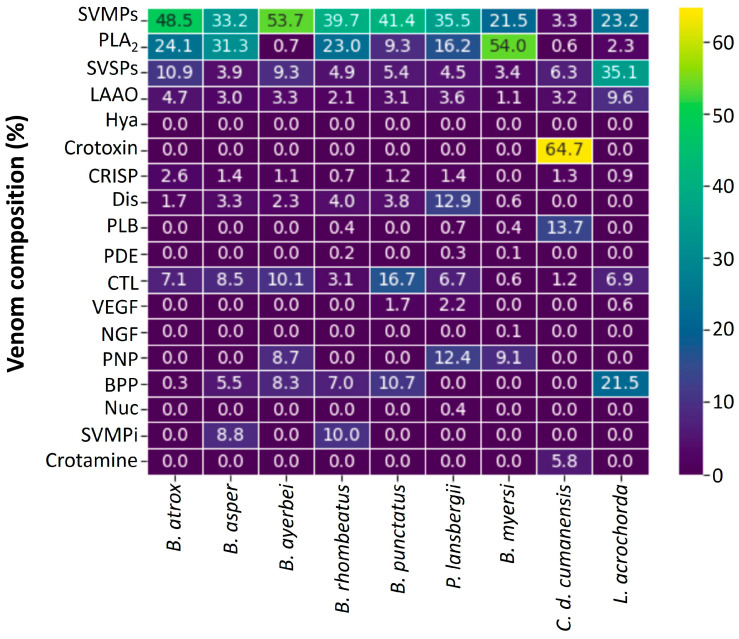
Quantitative venom proteomes from Viperidae family. Venom localities: *B. atrox* (Meta) [17], *B. asper* (Cauca) [15,16], *B. ayerbei* (Cauca) [16], *B. rhombeatus* (Cauca) [15], *B. punctatus* (Antioquia) [18], *P. lansbergii* (Caribbean) [20], *B. myersi* (Valle del Cauca) [19], *C. d. cumanensis* (pool from Meta, Tolima, Cundinamarca, and Magdalena), *L. acrochorda* (pool from Antioquia and Chocó). Abbreviations for protein family names: PLA_2_s: phospholipase A_2_; SVMPs: metalloproteinase; LAAO: L-amino acid oxidase; CTL: C-type lectin/lectin-like; CRISP: cysteine-rich secretory protein; Dis: Disintegrins; SVSPs: serine proteinase; Nuc: nucleotidase; PDE: phosphodiesterase; Hya: hyaluronidase; NGF: nerve growth factor; PLB: phospholipase B; PNP: peptides and/or nonproteinaceous components; BPP: bradykinin-potentiating peptide. The unknown fractions were not considered in this graph.

**Figure 3 toxins-15-00658-f003:**
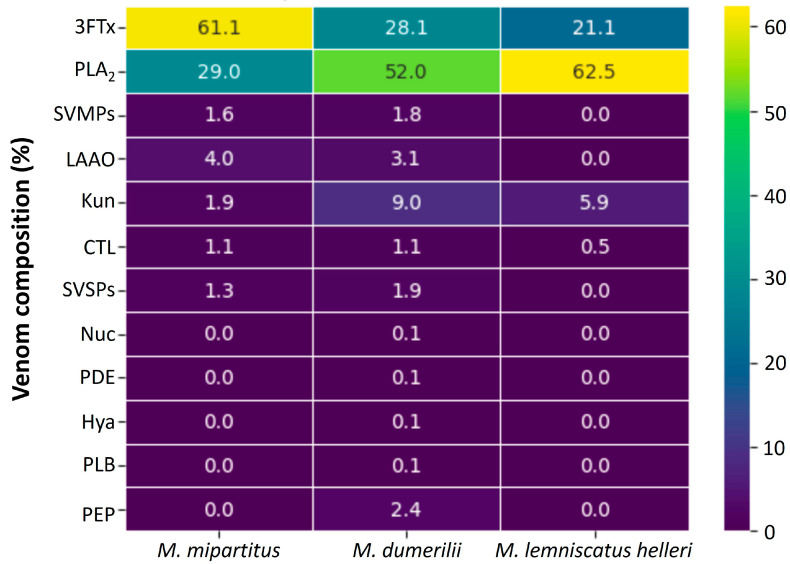
Quantitative venom proteomes from the Elapidae family. Venom localities: *M. mipartitus* (Antioquia) [29], *M. dumerilii* (Antioquia) [30], and *M. lemniscatus helleri* (Amazonas) [31]. Abbreviations for protein family names: 3FTx: three-finger toxins; PLA_2_s: phospholipase A_2_; SVMPs: metalloproteinase; LAAO: L-amino acid oxidase; CTL: C-type lectin/lectin-like; SVSPs: serine proteinase; Nuc: nucleotidase; PDE: phosphodiesterase; Hya: hyaluronidase; Kun: Kunitz-type inhibitors; PLB: phospholipase B; PNP: peptides and/or nonproteinaceous components. The unknown fractions were not considered in this graph.

**Figure 4 toxins-15-00658-f004:**
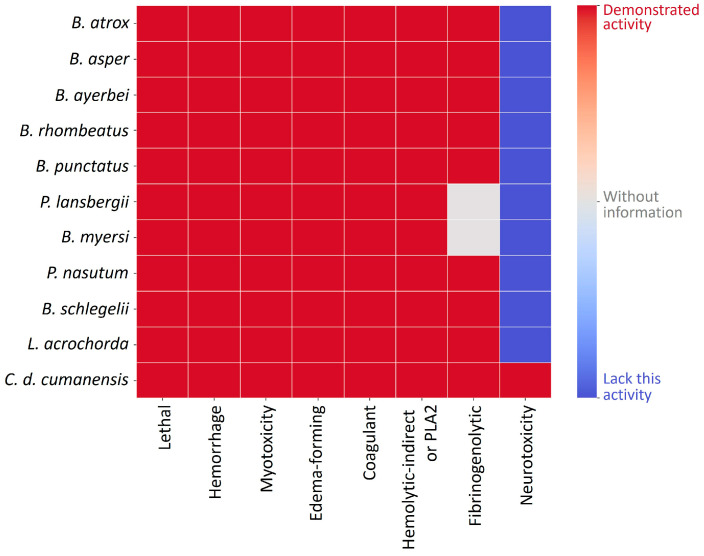
Biological activities reported for Colombian snake venoms from the Viperidae family. *B. atrox* (Meta) [33]; *B. asper* (Cauca and Antioquia) [16,34]; *B. ayerbei* (Cauca) [16]; *B. rhombeatus* (Cauca) [16]; *B. punctatus* (Antioquia) [18]; *P. lansbergii* (Atlántico) [20]; *B. myersi* (Valle del Cauca) [19]; *P. nasutum* (Antioquia) [34]; *B. schlegelii* (Antioquia) [34,37]; *L. acrochorda* (Antioquia) [34]; *C. d. cumanensis* (Meta) [34].

**Figure 5 toxins-15-00658-f005:**
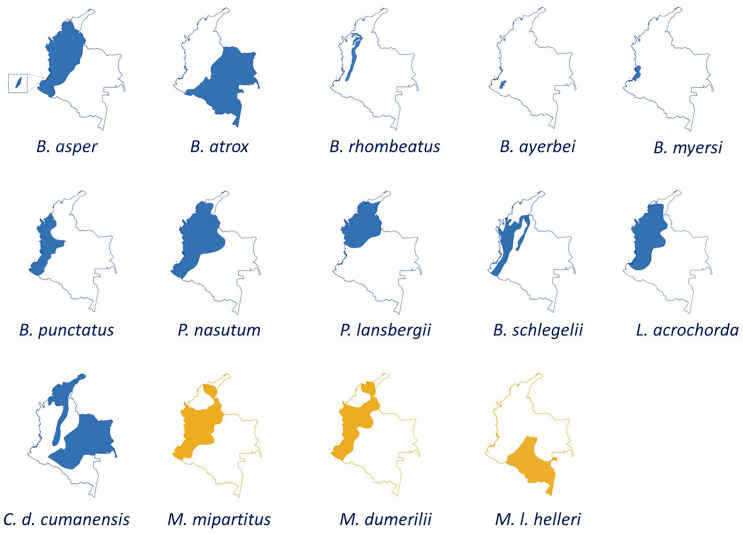
Geographical distribution of snake venoms of Viperidae (blue) and Elapidae (yellow) families described in this systematic review.

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
