# Peer review of "Knowledge about Snake Venoms and Toxins from Colombia: A Systematic Review"

_toxins, 2023, doi:10.3390/toxins15110658_

Round 1
Reviewer 1 Report
Comments and Suggestions for Authors
The article is very good. Provides relevant information for studies on snake venom from Colombia. Furthermore, it can help researchers who work in the area of antivenoms not only in Colombia but in other countries.
Results and discussion
Line 82: Check the table values.
Figure 2: Review the caption including some toxins that are missing. Example: Dis.
Line 347 and 349: Standardize Acetylcholine (Ach or ACh).
Author Response
We would like to thank the reviewers for their careful reading of the manuscript and their detailed criticisms, which have helped us to improve it.

Reviewer 2 Report
Comments and Suggestions for Authors
This manuscript provides a systematic review of species of snake found in Colombia and revision of the toxins found in their venom. The perspective a synthesis of the current knowledge however I would like to see the manuscript do a little more than recapitulate what we know – such as where are the gaps and where should research focus etc.
Specific issue to be addressed includes grammar and spelling – I appreciate that the authors may not have English is their native language, the manuscript would benefit in finding a native speaker of English to review the paper prior to acceptance. I have provided some examples below.
Use of redundant words making the paper harder to read e.g., “huge mountain” mountains are huge by nature so no need to specify that, “This review compilates” the grammar would be this paper is a compilation of information or “here we compile the knowledge”.
I think the review would be more useful to link the species toxins & symptoms of human envenomation. For example, L 223 symptoms observed – what are the symptoms? The review should delve deeper on these levels at present it is a broad level without any detail and in L 240 what is the implications of lower molecular masses in snakes as they age – is there a correlation between symptoms fatalities based on snake age? Is this a gap in the knowledge that needs to be addressed?
L 27 states very broadly what venom contains but it is not informative – best to provide examples of each “molecule” or “ion”.
L32 This is quite an old reference for number of snake species and would be more like 3500-4000 – please update with a more current number for snakes.
In all figure legends ensure the species name has been italicised
L141 & 146 ensure there is a full stop at the end spp e.g. “spp.” Check throughout the manuscript
Figure 4. Is it possible to put the LD50 or measure of toxicity to form of comparison of species within the boxes?
L189 In vivo should be lower case in vivo.
Comments on the Quality of English LanguageThe English and a lot of wording can be corrected to make it a more concise enjoyable article for readers
Author Response

(The authors gave the same response as above.)

Reviewer 3 Report
Comments and Suggestions for Authors
The authors present a well-executed systematic review of information concerning the venoms of snakes found in Columbia. Using standard PRISMA guidelines and methodology, 52 works were identified that addressed the proteomics and specific activities (LD50, neurotoxicity, etc.). The authors further subdivided these broad categories by family of snake (Viperidae, Elapidae) and specific species. Figures 2-4 are very informative and make the complex and comprehensive information facile to understand.
The authors then proceed to focus on seven genus/specific species that are medically important and have the most scientific information available. This was a clinically relevant transition that emphasized the toxicity was informative and should be of good use to the readership.
The last substantive section of the review re-collated the information previously presented to focus on antibacterial, antiplasmodial, and antitumoral properties of venoms from Columbia. Again, this section was a good clinical distillation that focused on the potential medicinal uses of Columbian snake venoms.
I have no important criticisms to offer concerning this fine work. I would like to ask the authors to consider providing a map of Columbia with regions identified by family and/or genus of the snakes discussed. Given the isolation of some of the islands mentioned and other specific regions discussed, it would be useful to the readership to know the geographical distribution.
Author Response

(The authors gave the same response as above.)

Round 2
Reviewer 2 Report
Comments and Suggestions for Authors
Recommendation to accept after considerations and response to review thus improving the manuscript.